# Effect of Denoising and Deblurring ^18^F-Fluorodeoxyglucose Positron Emission Tomography Images on a Deep Learning Model’s Classification Performance for Alzheimer’s Disease

**DOI:** 10.3390/metabo12030231

**Published:** 2022-03-07

**Authors:** Min-Hee Lee, Chang-Soo Yun, Kyuseok Kim, Youngjin Lee

**Affiliations:** 1Institute of Human Genomic Study, College of Medicine, Korea University Ansan Hospital, 123 Jeokgeum-ro, Danwon-gu, Ansan 15355, Korea; minheelee@korea.ac.kr; 2Department of Radiation Convergence Engineering, College of Software and Digital Healthcare Convergence, Yonsei University, 1, Yeonsedae-gil, Heungeop-myeon, Wonju 26493, Korea; ycs0709@yonsei.ac.kr; 3Department of Integrative Medicine, Major in Digital Healthcare, Yonsei University College of Medicine, Unju-ro, Gangman-gu, Seoul 06229, Korea; 4Department of Radiological Science, College of Health Science, Gachon University, 191, Hambakmoero, Yeonsu-gu, Incheon 21936, Korea

**Keywords:** ^18^F-FDG PET, deep convolutional neural network, Alzheimer’s disease

## Abstract

Alzheimer’s disease (AD) is the most common progressive neurodegenerative disease. ^18^F-fluorodeoxyglucose positron emission tomography (^18^F-FDG PET) is widely used to predict AD using a deep learning model. However, the effects of noise and blurring on ^18^F-FDG PET images were not considered. The performance of a classification model trained using raw, deblurred (by the fast total variation deblurring method), or denoised (by the median modified Wiener filter) ^18^F-FDG PET images without or with cropping around the limbic system area using a 3D deep convolutional neural network was investigated. The classification model trained using denoised whole-brain ^18^F-FDG PET images achieved classification performance (0.75/0.65/0.79/0.39 for sensitivity/specificity/F1-score/Matthews correlation coefficient (MCC), respectively) higher than that with raw and deblurred ^18^F-FDG PET images. The classification model trained using cropped raw ^18^F-FDG PET images achieved higher performance (0.78/0.63/0.81/0.40 for sensitivity/specificity/F1-score/MCC) than the whole-brain ^18^F-FDG PET images (0.72/0.32/0.71/0.10 for sensitivity/specificity/F1-score/MCC, respectively). The ^18^F-FDG PET image deblurring and cropping (0.89/0.67/0.88/0.57 for sensitivity/specificity/F1-score/MCC) procedures were the most helpful for improving performance. For this model, the right middle frontal, middle temporal, insula, and hippocampus areas were the most predictive of AD using the class activation map. Our findings demonstrate that ^18^F-FDG PET image preprocessing and cropping improves the explainability and potential clinical applicability of deep learning models.

## 1. Introduction

Alzheimer’s disease (AD) is a progressive neurodegenerative disease characterized by cognitive decline and memory loss [1], and it is the most common cause of dementia, which causes disability and dependency in older people worldwide [2]. Cognitive decline may be associated with metabolic and neurotransmitter activities in the brain [3]. These changes in AD may start several years before the onset of clinical symptoms [4,5]. Thus, early detection of AD is important, as early treatment of this disease may delay its progression [2]. To acquire information on pathological processes related to AD, ^18^F-fluorodeoxyglucose (^18^F-FDG) positron emission tomography (PET), which reflects the glucose metabolism of cerebral neurons, is widely used [3,6]. However, since the earliest symptoms of AD, such as short-term memory loss, are confused with symptoms resulting from aging, stress, or other brain disorders, it remains challenging to recognize AD before the manifestation of severe cognitive impairment with typical neuroimaging signs [7]. Additionally, the reliance on the interpretation of qualitative readings by specialists in order to recognize AD patterns is an issue in the clinical application of brain ^18^F-FDG PET [8]. Thus, a recognized approach to detect AD in the early stages is urgently needed.

In nuclear medicine, recent developments in artificial intelligence (AI) methodology allow the extraction of brain metabolic activity features related to neurodegenerative disorders, and it has been utilized to classify AD and normal conditions using ^18^F-FDG PET images [6,9,10]. Ding et al. reported that the classification accuracy for three groups (cognitively normal control, mild cognitive impairment (MCI), and AD groups) using a classification model of ^18^F-FDG PET images implemented with a convolutional neural network (CNN) technique was higher than that of radiologists’ performance [6]. Zhou et al. proposed a deep learning model to assist in the diagnosis of conversion to AD from MCI [9]. Although previous studies provided an effective deep learning model to predict AD, the effects of noise and blurring of the ^18^F-FDG PET images in the classification model were not considered. Because adversarial images can lead to misclassification, deep learning models are susceptible to image noise or blurring. This consequently reduces the performance and explainability of the deep learning model [11,12].

The origin of the PET signal is the radioactivity decay of the labelled tracer, and the underlying decay process follows Poisson statistics [13]. Therefore, the noise of the PET image is determined by the number of registered counts, and the positron range and motion contribute to the blurring of PET images [13]. Because the unavoidable noise and blurring of PET images reduce the signal-to-noise ratio and spatial resolution, correction and the application of a filter are required to accurately recognize AD. Software or image processing algorithms to reduce noise and blurring in PET images of AD are constantly being developed. However, few studies have measured the change in accuracy of AD classification according to the presence or absence of an image improvement algorithm based on deep learning technology. Furthermore, because irrelevant and redundant features degrade the accuracy and efficiency of a classification model [14,15], we compared the performance of the deep learning model using input features that included the whole brain or just the regions of interest. 

This study aimed to investigate: (1) whether applying the denoising and deblurring method to ^18^F-FDG PET images improves the performance of AD classification, and (2) whether ^18^F-FDG PET image cropping improves the performance of AD classification using a modified deep learning model from 3D-ResNet, which has been recently described as a powerful prediction model for 3D medical images [16]. We also investigated the explainability difference using a class activation map [17] between deep learning outputs of raw images and denoised or deblurred images. Our primary hypothesis is that applying the preprocessing (i.e., denoising or deblurring method and cropping) to the input features can improve the performance of the deep learning model. In the present study, we attempted to improve the deep learning-based assessment of ^18^F-FDG PET images, where current approaches are suboptimal for controlling ^18^F-FDG PET image distortions, by systematically evaluating whether denoising and deblurring ^18^F-FDG PET images can improve the performance of deep learning models. Ultimately, such an application could be an effective tool for accurately classifying AD patients and cognitively normal controls.

## 2. Results

### 2.1. Demographic Characteristics and Clinical Assessments

Demographic data including age, sex, education, and neuropsychological cognitive assessment tests such as the Mini-Mental State Examination (MMSE), Clinical Dementia Rating (CDR) scale, and apolipoprotein E (APOE) ε4 genotyping characteristics are shown in Table 1. There were no differences in age, sex, or education level. However, the AD group was more likely to carry the APOE ε4 allele (*p* < 0.001) and have lower cognitive test performance results (i.e., MMSE and CDR, *p* < 0.001).

### 2.2. Classification Performance

Figure 1 shows the convergence curves of loss function for the 3D deep CNN during the training and testing process based on raw, deblurred, or denoised whole-brain ^18^F-FDG PET images and ^18^F-FDG PET images cropped around the limbic system area. The training and testing loss of 3D deep CNN was converged at the 20th epoch regardless of data processing. 

The performance comparisons of the individual classification models trained using raw, denoised, or deblurred whole-brain ^18^F-FDG PET images to classify AD and cognitively normal conditions are summarized in Table 2. The classification model trained using deblurred whole-brain ^18^F-FDG PET images with a σ_b_ of 2 achieved the highest average sensitivity of 0.91, but not the highest specificity, accuracy, F1-score, and Matthews correlation coefficient (MCC). The classification model trained using denoised whole-brain ^18^F-FDG PET images achieved higher average specificity, accuracy, F1-score, and MCC than raw and deblurred ^18^F-FDG PET images. Although the deblurring method was helpful for improving classification sensitivity, the denoising method (σ_n_ = 3, 0.75/0.65/0.72/0.79/0.39 and σ_n_ = 5, 0.85/0.48/0.74/0.82/0.35 for sensitivity/specificity/accuracy/F1-score/MCC, respectively) was more effective than the deblurring method (σ_b_ = 1, 0.83/0.30/0.67/0.78/0.14 and σ_b_ = 2, 0.91/0.25/0.71/0.81/0.21) for the classification of AD patients and cognitively normal controls using whole-brain ^18^F-FDG PET images.

The performance comparisons of individual classification models trained using raw, denoised, or deblurred ^18^F-FDG PET images that were cropped around the limbic system area to classify AD and normal conditions are summarized in Table 3. The classification model trained using raw ^18^F-FDG PET images cropped around the limbic system area achieved higher classification performance (0.78/0.63/0.74/0.81/0.40 for sensitivity/specificity/accuracy/F1-score/MCC, respectively) than the whole-brain ^18^F-FDG PET images (0.72/0.32/0.60/0.71/0.10). This may mean that the ^18^F-FDG PET image cropping is helpful for improving classification performance. The deblurring method with a σ_b_ of 2 (0.89/0.67/0.82/0.88/0.57) was most helpful for improving the model’s classification performance using ^18^F-FDG PET images cropped around the limbic system area.

### 2.3. Activation Maps Associated with the Classification of Alzheimer’s Disease and Cognitively Normal Controls

Figure 2 presents the activation maps showing important brain regions that the 3D deep CNN learned as the most predictive of AD. Each map was averaged over the participants. An image intensity with a greater weight was more predictive of AD. A wide range of non-brain regions were included as the most predictive brain regions, regardless of image processing performed in the classification model trained using whole-brain ^18^F-FDG PET images. However, non-brain regions were not included as the most predictive brain regions in the classification model trained using denoised or deblurred ^18^F-FDG PET images, which had been cropped around the limbic system area.

In the classification model trained using deblurred and cropped ^18^F-FDG PET images, the right middle frontal gyrus, insula, middle temporal gyrus, and hippocampus area were the most predictive of AD. Moreover, the right thalamus, left caudate, and bilateral putamen areas were the most predictive of AD in the classification model trained using denoised and cropped ^18^F-FDG PET images.

## 3. Discussion

The present study provides three major findings supporting the performance of the AD classification model trained using preprocessed ^18^F-FDG PET images cropped around the limbic system area over other models (i.e., the classification model trained using raw and preprocessed whole-brain ^18^F-FDG PET images and trained using raw ^18^F-FDG PET images cropped around the limbic system area). First, denoising and deblurring methods are helpful for improving classification performance indicators, including sensitivity, specificity, accuracy, F1-score, and MCC. Second, cropping whole-brain ^18^F-FDG PET images around the limbic system area improved the model’s classification performance. Third, preprocessing the input improved the explainability of the model’s classification using a class activation map, which improved the inference process and clinical interpretation.

This study was motivated by the hypothesis that deep learning models are influenced by image quality, which varies depending on several factors. The performance of a deep learning model trained by an input dataset with noisy images could worsen when noisy features are captured [12,18,19]. Therefore, the input image quality must be considered to improve the performance of a deep learning model [12,18,19]. The quality of PET images is influenced by various factors, including imaging system hardware, non-collinearity of the emitted photon pairs, intercrystal scatter, and crystal penetration [20,21]. The development of denoising and deblurring methods for PET imaging remains an important research avenue to facilitate clinical decision-making and interpretation [22,23]. In the present study, ^18^F-FDG PET images were obtained from 50 PET centers having nine different scanner models. In spite of using a standardized imaging protocol, there is inter-scanner variability in ^18^F-FDG-PET images due to differences in scintillator materials, scintillator size, image reconstruction algorithm, image size, and slice thickness [24] (Table 4). For this reason, the noise type or level of the PET images might have varied. In the present study, these unavoidable and various noisy characteristics of ^18^F-FDG PET images practically deteriorated the performance of the classification model of AD, as the classification model trained using raw ^18^F-FDG PET images learned features of irrelevant or non-brain regions (see Figure 2). After preprocessing (i.e., deblurring and denoising), the performance of the classification model was significantly improved (MCC of 0.10 before preprocessing vs. MCC of up to 0.39 after preprocessing). Nevertheless, few non-brain regions remained important for classifying AD patients and cognitively normal controls. This may still hamper clinical decision-making and interpretation. Therefore, this suggests a need for minimizing the confounders of the classification model.

Image cropping is advantageous, as it elucidates a more focused region of interest to facilitate feature extraction from images, and it allows for the reduction of noisy image components [25]. A previous study verified that a cropping-based image classification model improved classification accuracy [25]. In the present study, ^18^F-FDG PET images were cropped around the limbic system area to focus on this area and reduce noisy image components (see Figure 3). After ^18^F-FDG PET image cropping, the performance of the classification model was conspicuously improved (MCC = 0.1 before cropping vs. MCC = 0.4 after cropping raw images). After image preprocessing and cropping, the performance of the classification model was significantly improved (MCC = 0.1 before both image preprocessing and cropping vs. MCC = up to 0.57 after both image preprocessing and cropping). Furthermore, non-brain regions were not important for classifying AD patients and cognitively normal controls. These findings suggest that although either image preprocessing or cropping could improve the performance of the classification model, image preprocessing and cropping improved the performance and clinical applicability of the classification model.

Previous neuroimaging studies have demonstrated that AD is associated with brain structural or functional alterations in a wide range of brain areas, including the middle frontal gyrus, middle temporal gyrus, and limbic system areas such as the hippocampus, insula, thalamus, putamen, and caudate [26,27,28,29,30,31,32]. The middle frontal and middle temporal gyri are related to verbal short-term memory performance [31], and the limbic system area is known to be functionally related to cognition, autonomic, emotional, and sensory processes [26,27,28]. Further, AD patients showed distinct patterns of cerebral glucose metabolism due to loss of synapses and neuropil, as well as functional impairment of the neurons in these areas [29,32]. These patterns allow for the differentiation of AD from cognitively normal controls using ^18^F-FDG PET images [33]. In the present study, the most predictive brain regions in the class activation map were obtained from the classification model trained using pre-processed ^18^F-FDG PET images that were cropped around the limbic system area were largely consistent with previous studies. Our findings provide evidence that preprocessing the input features facilitates clinical decision-making and interpretation of the classification model based on ^18^F-FDG PET images.

The present study had several limitations. First, because we focused on the effects of preprocessing the input data on the performance of the AD classification model, other deep learning models were not considered for evaluating the classification performance. Although this is beyond the scope of the current study, it remains an important line of inquiry for future research. In addition, we did not consider regional harmonization and intensity normalization using state-of-the-art methods such as removal of artificial voxel effect by linear regression (RAVEL) and combining batches (ComBat) to reduce scanner effects and improve reproducibility of image intensities [34,35]. Further studies applying state-of-the-art methods to our proposed methods are needed. Nevertheless, we attempted to employ widely used models such as the 3D deep CNN model for training ^18^F-FDG PET images, a fast total variation (TV)-*l*_1_ deblurring method, and the median modified Wiener filter (MMWF) for denoising. Second, the sample size (cognitively normal controls, n = 155, AD patients, n = 66) was not large enough to achieve an ultimate AD classification model via the determination of the heterogeneity of the ^18^F-FDG PET image data using the present deep learning technique, or to avoid over- or under-estimation of classification model performance due to chance. To minimize this limitation, we used 3-fold cross-validation [36]. Third, in the present study, since MCI is an unstable diagnosis in that its accuracy is dependent on the source of the patients, the specific criteria used, the methods employed, and the length of follow-up [37], we did not consider classifying MCI. However, this remains an important line of inquiry for future research. Finally, delicate work is required in image restoration and input value for the deep learning model. In this study, a two-dimensional (2D) point-spread function (PSF) was used to improve the reliability of the results, because the PET reconstruction method was not unified (see Table 4). Since we used a 3D deep learning model, it is necessary to check the difference in results between the input image using the 2D and 3D PSF based on the PET image with the reconstruction algorithm matched. Moreover, results according to various input types (e.g., multiple input of deblurred and denoised image, deblurred image after denoising, and vice versa) of the deep learning model were derived; however, significant results and correlation were not identified. To overcome this problem, further research is being conducted.

## 4. Materials and Methods

### 4.1. Data Acquisition and Preprocessing

All data used in this study were obtained from the Alzheimer’s Disease Neuroimaging Initiative (ADNI) database (https://adni.loni.usc.edu, accessed on 31 October 2021). The ADNI was launched in 2003, with the primary goal of testing whether serial magnetic resonance imaging (MRI), PET, other biological dementia markers, and structured clinical and neuropsychological assessments could be combined to measure the progression of MCI and early AD. 

In the present study, we included 155 cognitively healthy controls (age, 75.31 ± 6.57 years; men, 52.26%) and 66 patients (age, 74.44 ± 8.39 years; men, 50.00%) clinically diagnosed with AD from ADNI Phases 1, 2, or Go who underwent APOE testing, cognitive assessments, and ^18^F-FDG PET scans. Diagnostic criteria were available on the ADNI website. Briefly, participants with AD met the National Institute of Neurological Disorders and Stroke Alzheimer’s Disease and Related Disorder Association (NINDS-ADRDA) criteria for probable AD [38].

The ^18^F-FDG PET images with six 5-min frames were acquired beginning 30 min after injection of 5.0 ± 0.5 mCi (i.e., 185 MBq) of ^18^F-FDG using a Siemens scanner. Detailed information on FDG-PET image acquisition and preprocessing is available on the ADNI website (http://adni.loni.usc.edu/methods/documents/, accessed on 31 October 2021). All ^18^F-FDG PET images were spatially normalized to a standard space using SPM12 (https://www.fil.ion.ucl.ac.uk/spm/software/spm12/, accessed on 24 November 2021). The processed images had an image size of 91×109×91 and a voxel size of 2×2×2  mm^3^. Lastly, each 3D ^18^F-FDG PET image was cropped around the limbic system area (i.e., including the hippocampus, amygdala, thalamus, and putamen, Figure 3), a known metabolic dysfunction-associated area in AD patients [39], using an automated anatomical labeling atlas 2 template [40] to investigate the effect of reducing irrelevant and redundant features on classification performance.

All individuals were randomly assigned into two groups: a training set (cognitively normal controls, n = 109; AD patients, n = 46) to implement AD classification using the deep learning model of ^18^F-FDG PET images and testing set (cognitively normal controls, n = 46; AD patients, n = 20) to confirm the convergence of the AD classification and to reproduce the accuracy of the AD classification.

### 4.2. Cognitive Assessment

The MMSE is an instrument for cognitive assessment, with raw scores ranging from 0 to 30 [41]. The CDR [42,43] is a clinician-rated staging cognitive assessment method that requires an individual severity rating in each of the following six domains: (1) memory, (2) orientation, (3) judgment and problem-solving; (4) community affairs; (5) home and hobbies; and (6) personal care. An overall global CDR score that indicates five stages of cognitive dysfunction (0 = no dementia, 0.5 = very mild dementia, 1 = mild dementia, 2 = moderate dementia, and 3 = severe dementia) may be calculated through the CDR scoring algorithm [43]. Alternatively, the Sum of Boxes scoring approach (CDRSB) yields scores from 0 to 18 by summing the CDR domain scores [44]. 

### 4.3. Image Restoration

Generally, PET image restoration should consider both the sinogram domain and the image (voxel) domain. Multiplicative image degradation occurs because of the inherent performance of the detector, which is affected in the sinogram domain, and the positron range in the image domain during reconstruction [20]. However, only image degradation in the image domain is considered based on the limitations of the information provided by the ADNI. The degradation model in the image domain [45] can be written as shown in Equation (1):(1)g=psf⊗⊗f+N,
where *g* is the degraded image, *psf* is the PSF, which is the amount of blurring that degrades from the clean image, *f*, and *N* is the noise component. Here, ⊗⊗ represents a 2D convolution. Here, the PSF is defined as:(2)$$psf\left(m,n\right)=\exp\left(-\frac{m^{2}+n^{2}}{2\sigma_{b}{}^{2}}\right),$$
where $\sigma_{b}$ is standard deviation of the Gaussian kernel, and *m* and *n* are discrete indices in the image domain. When deblurring and denoising are implemented, perfect restoration may be difficult to achieve. In deblurring, although the resolution and sharpness of the image are improved, the noise component in the high-frequency region may also be amplified. Therefore, we employed a fast TV-*l*_1_ deblurring method [46,47], which is well known to improve the resolution while suppressing noise amplification. The object function is expressed as follows:(3)$$f^{*}=\underset{f\in Q}{\mathrm{argmin}}\left(||psf⊗⊗f-g||+\lambda ||\nabla f||\right),$$
where the object function consists of the fidelity term (||psf⊗⊗f−g||) and the penalty term (||∇f||). A balancing parameter, λ, is implemented to increase the signal-to-noise ratio (i.e., *λ* = 0.01 was used in this study). The aim of this problem is to find the $f^{*}$ that minimizes the object function without additional artifacts (e.g., ringing artifact [48]). Here, the solution method in Equation (2) uses a half-quadratic splitting approach [49].

Nevertheless, conventional denoising methods have a limitation in reducing sharpness with noise reduction. Cannistraci et al. [50] introduced the median modified Wiener filter (MMWF), and Park et al. [51] showed improved image performance using the MMWF in gamma camera images. It was confirmed that the MMWF is effective in suppressing the noise component while preserving the outline of the object as much as possible. The MMWF is represented as follows:(4)$$b_{mmwf}\left(n,m\right)=\bar{\mu }+\frac{\sigma_{n}{}^{2}-\gamma^{2}}{\sigma_{n}{}^{2}}\cdot \left(\Omega \left(n,m\right)-\bar{\mu }\right),\;n,m\in \eta$$
where $\bar{\mu }$ and $\sigma_{n}{}^{2}$ are the local median mean and variance around each pixel, n-by-m is the size of the neighborhood area η in the mask, Ω(n,m) represents each pixel in η, and $\gamma^{2}$ is the noise variance. Here, we used the average of all the local variances.

### 4.4. Proposed Framework

Figure 4 shows the simplified schematic illustration for the prediction of cognitively normal and AD patients using the proposed framework. In brief, a ^18^F-FDG PET image for training implemented the image restoration using Equation (2) or Equation (3). In some cases, image restoration could not be performed to evaluate the prediction performance due to variations in the image quality. Here, the $\sigma_{b}$ of the *psf* in Equation (2) was 2 and the size of Ω(n,m) was 5 × 5. This processing was applied to all slices with the same parameters, including the σ and Ω(n,m) size. Following this, each model was trained using the ^18^F-FDG PET dataset, which was predetermined using the image restoration components. Finally, cognitively normal or AD conditions could be predicted using a trained model with the same pre-training conditions. Moreover, a class activation map was also deduced to interpret the prediction decisions.

Figure 5 shows the proposed 3D deep CNN architecture, which consists of an input layer, hidden layers, and an output layer. In this network, a small block is composed of a convolution layer, batch normalization (BN) layer, and an activation layer such as a rectified linear unit (ReLU). Additionally, residual blocks have branches such as convolution and BN layers, which use the concept-of-ensemble approach [52]. The loss function was implemented by means of cross entropy, and the adaptive momentum estimation optimizer was used to update the parameters in the back-propagation. The batch size was 4, the number of epochs was 20, and the initial learning rate was 10^−4^. The architecture of the network is summarized in Table 5.

We implemented the proposed scheme using a normal workstation (OS: Windows 10, CPU: 2.13 GHz, RAM: 64 GB, GPU: Titan Xp, 12 GB) and the software language employed for the image processing and deep learning was MATLAB^™^ (R2020b, MathWorks, Natick, MA, USA). Of the total data from 221 individuals, 70% (cognitively normal controls, n = 109; AD patients, n = 46) was used for training, and the remaining share (cognitively normal controls, n = 46; AD patients, n = 20) was used for testing.

### 4.5. Statistical Analysis

General and clinical characteristics were compared between the groups using the two-sample *t*-test or chi-square test.

The performance of the deep learning model was evaluated using the sensitivity, which is a measure of the true positive rate; the specificity, a measure of the true negative rate; and the accuracy, which is a measure of the proportion of correct classifications calculated from a confusion matrix for a two-class classification problem [53]. In the present study, since the participants were mainly cognitively normal controls (controls, 70%; AD patients, 30%; i.e., imbalanced data), the F1-scores and MCCs, which are effective measures for imbalanced datasets, were also used [54]. Moreover, since our data sample was not sufficient to represent the whole population, and since the performance of the classification model might be due to chance, we employed 3-fold cross-validation, a resampling procedure that could be used to evaluate our 3D deep-CNN model [36]. 

## 5. Conclusions

In summary, the present study provides data that explain the importance of preprocessing input data for training classification models using deep learning methods. Furthermore, our study demonstrates that preprocessing the input data improves the explainability and potential clinical applicability of the deep learning method. Therefore, our approach may encourage future studies to develop a classification model trained using ^18^F-FDG PET images using a deep learning method.

## Figures and Tables

**Figure 1 metabolites-12-00231-f001:**
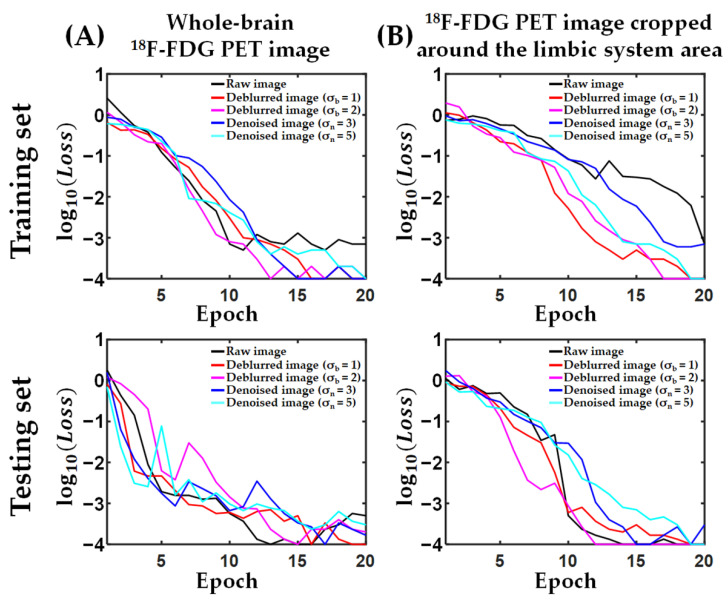
Convergence curves of a 3D deep-convolutional neural network trained using raw, deblurred, or denoised whole-brain ^18^F-fluorodeoxyglucose positron emission tomography (^18^F-FDG PET) images (**A**) and ^18^F-FDG PET images cropped around the limbic system area (**B**). A semi-logarithmic line plot has a logarithmic scale of loss on the *y*-axis.

**Figure 2 metabolites-12-00231-f002:**
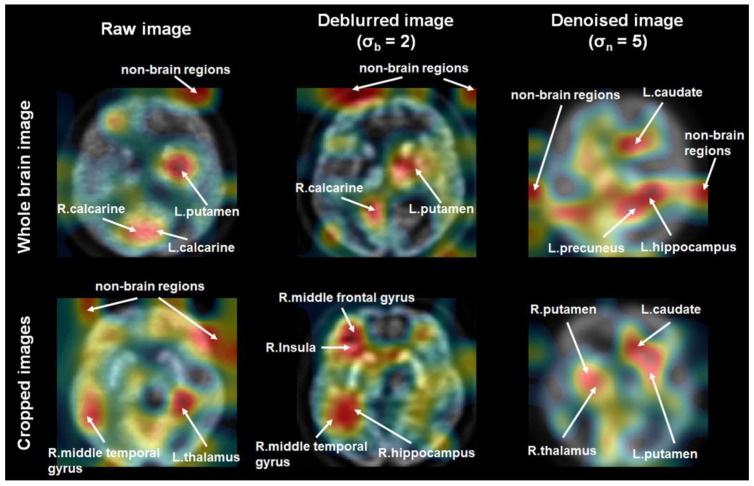
Activation maps showing brain regions learned by the 3D deep- convolutional neural network model as the most predictive of Alzheimer’s disease. Higher weights (red) indicate that the image intensity was more predictive of Alzheimer’s disease. Abbreviations: L, Left; R, Right.

**Figure 3 metabolites-12-00231-f003:**
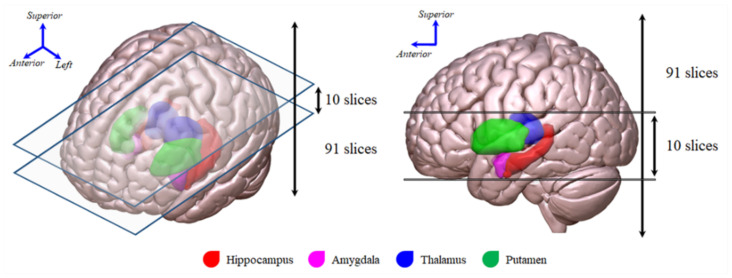
Slices of interest, including the hippocampus, amygdala, thalamus, and putamen, that were used for the ^18^F-FDG PET image cropping.

**Figure 4 metabolites-12-00231-f004:**
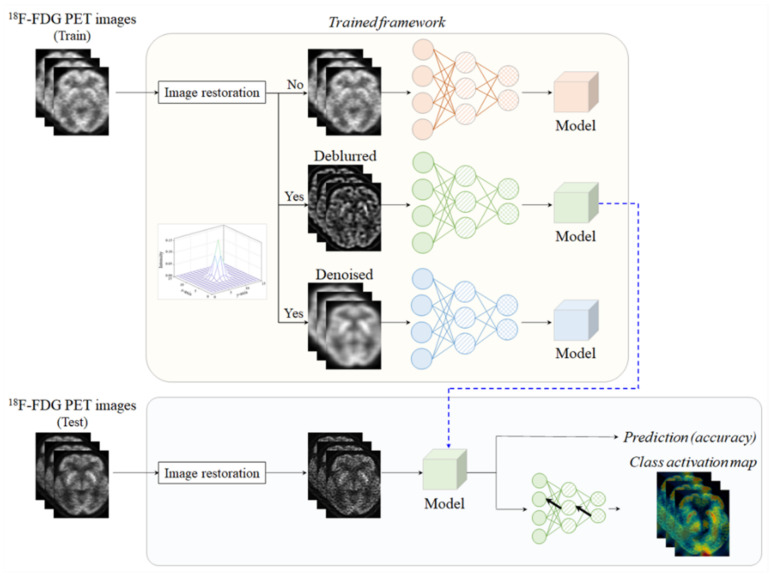
Simplified diagram representing the proposed prediction process incorporating image restoration using the deep convolutional neural network trained using ^18^F-FDG PET images.

**Figure 5 metabolites-12-00231-f005:**
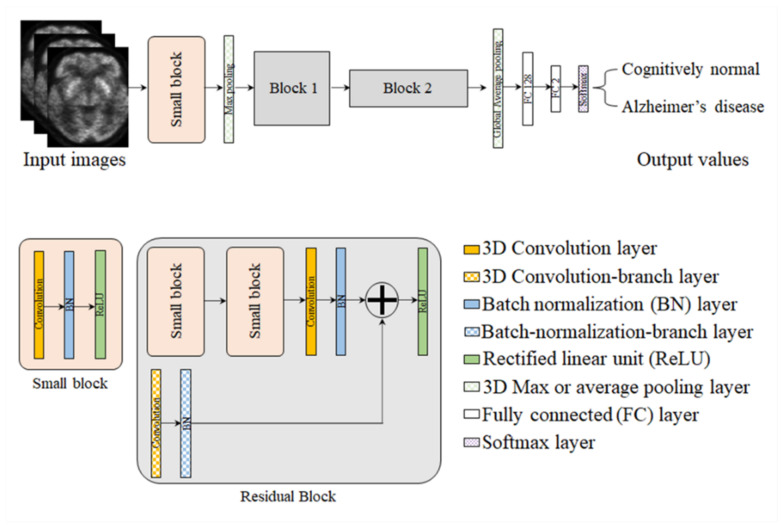
The structure of the proposed network. Most network parameters used in this study are consistent with the notations in this figure.

**Table 1 metabolites-12-00231-t001:** Demographic and clinical assessments.

	Cognitively Normal Controls (n = 155)	Patients with Alzheimer’s Disease (n = 66)	*p*
Age	75.31 ± 6.57	74.44 ± 8.39	0.412
Male sex	81 (52.26)	33 (50.00)	0.759
Education	16.17 ± 2.89	15.35 ± 2.90	0.054
APOE ε4, carriers	43 (27.74)	45 (68.18)	<0.001
MMSE score	29.03 ± 1.21	23.26 ± 2.15	<0.001
Global CDR score	0.00 ± 0.00	0.80 ± 0.29	<0.001
CDR Sum of Boxes	0.04 ± 0.15	4.53 ± 1.67	<0.001
Memory	0.00 ± 0.00	1.05 ± 0.40	<0.001
Orientation	0.00 ± 0.00	0.89 ± 0.40	<0.001
Judgment	0.04 ± 0.13	0.87 ± 0.34	<0.001
Community affairs	0.01 ± 0.08	0.76 ± 0.49	<0.001
Hobbies	0.00 ± 0.00	0.73 ± 0.51	<0.001
Personal care	0.00 ± 0.00	0.23 ± 0.42	<0.001

Abbreviations: APOE, apolipoprotein E; MMSE, Mini-Mental State Examination; CDR, Clinical Dementia Rating scale.

**Table 2 metabolites-12-00231-t002:** Performance comparison of an individual classification model trained by raw, denoised, or deblurred whole-brain ^18^F-FDG PET images for the classification of Alzheimer’s disease and cognitively normal conditions.

	Sensitivity	Specificity	Accuracy	F1-Score	MCC
Raw ^18^F-FDG PET	0.72 ± 0.07	0.32 ± 0.08	0.60 ± 0.07	0.71 ± 0.05	0.10 ± 0.09
Deblurred ^18^F-FDG PET (σ_b_ = 1)	0.83 ± 0.07	0.30 ± 0.09	0.67 ± 0.03	0.78 ± 0.03	0.14 ± 0.04
Deblurred ^18^F-FDG PET (σ_b_ = 2)	0.91 ± 0.08	0.25 ± 0.15	0.71 ± 0.05	0.81 ± 0.04	0.21 ± 0.15
Denoised ^18^F-FDG PET (σ_n_ = 3)	0.75 ± 0.08	0.65 ± 0.05	0.72 ± 0.05	0.79 ± 0.05	0.39 ± 0.08
Denoised ^18^F-FDG PET (σ_n_ = 5)	0.85 ± 0.06	0.48 ± 0.13	0.74 ± 0.05	0.82 ± 0.04	0.35 ± 0.14

Data are presented as means ± standard deviations over cross-validation folds. Abbreviations: MCC, Matthews Correlation Coefficient; ^18^F-FDG PET, ^18^F-fluorodeoxyglucose positron emission tomography.

**Table 3 metabolites-12-00231-t003:** Performance comparison of an individual classification model trained using raw, denoised, or deblurred ^18^F-FDG PET images that were cropped around the limbic system area in order to classify Alzheimer’s disease and cognitively normal conditions.

	Sensitivity	Specificity	Accuracy	F1-Score	MCC
Raw ^18^F-FDG PET	0.78 ± 0.06	0.63 ± 0.13	0.74 ± 0.03	0.81 ± 0.03	0.40 ± 0.09
Deblurred ^18^F-FDG PET (σ_b_ = 1)	0.85 ± 0.06	0.68 ± 0.06	0.80 ± 0.05	0.85 ± 0.04	0.53 ± 0.10
Deblurred ^18^F-FDG PET (σ_b_ = 2)	0.89 ± 0.06	0.67 ± 0.10	0.82 ± 0.07	0.88 ± 0.05	0.57 ± 0.17
Denoised ^18^F-FDG PET (σ_n_ = 3)	0.83 ± 0.08	0.50 ± 0.13	0.73 ± 0.08	0.81 ± 0.05	0.34 ± 0.19
Denoised ^18^F-FDG PET (σ_n_ = 5)	0.88 ± 0.06	0.63 ± 0.08	0.80 ± 0.03	0.86 ± 0.02	0.53 ± 0.05

Data are presented as means ± standard deviations over cross-validation folds. Abbreviations: MCC, Matthews Correlation Coefficient; ^18^F-FDG PET, ^18^F-fluorodeoxyglucose positron emission tomography.

**Table 4 metabolites-12-00231-t004:** Scanner models and inter-scanner differences in the ^18^F-FDG-PET scans.

Scanner Model	Scintillator Materials	Scintillator Size (mm^3^)	Reconstruction Algorithm	Image Size	Slice Thickness (mm)
Siemens HRRT	LSO	2.1 × 2.1 × 20	OSEM-3D	256 × 256 × 207	1.2
Siemens HR+	BGO	4.05 × 4.39 × 30	FORE/OSEM-2D	128 × 128 × 63	2.4
Siemens Accel	LSO	6.45 × 6.45 × 25	FORE/OSEM-2D	128 × 128 × 47	3.4
Siemens Exact	BGO	6.75 × 6.75 × 20	FORE/OSEM-2D	128 × 128 × 47	3.4
Siemens SOMATOM Definition AS mCT	LSO	4.0 × 4.0 × 20	OSEM-3D	400 × 400 × 109	2.0
Siemens SOMATOM Definition AS mCT	LSO	4.0 × 4.0 × 20	OSEM-3D	400 × 400 × 81	2.0
Siemens Biograph 64	LSO	4.0 × 4.0 × 20	OSEM-3D	400 × 400 × 109	2.0
Siemens Biograph mCT 20	LSO	4.0 × 4.0 × 20	OSEM-3D	256 × 256 × 81	2.0
Siemens 1094	LSO	4.0 × 4.0 × 20	OSEM-3D	336 × 336 × 109	2.0

FORE, Fourier rebinning; OSEM, the ordered subsets expectation–maximization.

**Table 5 metabolites-12-00231-t005:** Architecture of the network.

Layer	Shape	Filter of Pooling	Stride/Padding
Input	128 × 128 × 79 × 1 or 128 × 12 × 810 × 1	-	-
Conv	64 × 64 × 40 × 64 or 64 × 64 × 5 × 64	7 × 7 × 7	2/3
BN		-	-
ReLU		-	-
Max pooling	32 × 32 × 20 × 64 or 32 × 32 × 3 × 64	3 × 3 × 3	2/1
Conv and Conv-branch	32 × 32 × 20 × 64 or 32 × 32 × 3 × 64	3 × 3 × 3	1/1
BN and BN-branch		-	-
ReLU		-	-
Conv and Conv-branch	16 × 16 × 10 × 128 or 16 × 16 × 2 × 128	3 × 3 × 3	1/1
BN and BN-branch		-	-
ReLU		-	-
Global average pooling	1 × 1 × 1 × 128	-	-
Fully connected-128	1 × 1 × 1 × 128	-	-
Fully connected-2	1 × 1 × 1 × 2	-	-
Softmax	1 × 1 × 1 × 2	-	-
Classification output	1 × 1 × 1 × 2	-	-

Abbreviations: Conv, convolution; BN, batch normalization; ReLU, rectified linear unit.

## Data Availability

Data used in the preparation of this article were obtained from the ADNI database (adni.loni.usc.edu, accessed on 31 October 2021). The ADNI was launched in 2003 as a public–private partnership led by Principal Investigator Michael W. Weiner, MD. The primary goal of the ADNI has been to test whether serial magnetic resonance imaging (MRI), positron emission tomography (PET), other biological markers, and clinical and neuropsychological assessments can be combined to measure the progression of mild cognitive impairment (MCI) and early Alzheimer’s disease (AD). For up-to-date information, see www.adni-info.org (accessed on 31 October 2021).

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
