# Peer review of "Effect of Denoising and Deblurring 18F-Fluorodeoxyglucose Positron Emission Tomography Images on a Deep Learning Model’s Classification Performance for Alzheimer’s Disease"

_metabolites, 2022, doi:10.3390/metabo12030231_

Round 1
Reviewer 1 Report
The authors present a study on how the preprocessing the PET images (denoising, deblurring, selection of the region-of-interest) that are used for a classification Neural Network improves the results.
It is well written and it can be read easily, nevertheless, there are several things that could be improved to make it more relevant to potential readers:
1 ) Fig. 1 - (And Line 109) The fact that the curves converge faster does not mean that the methods are better. A log scale in the y-axis in these figure, maybe reveal the behaviour at large number of epochs. Furthermore, the loss function showed is not clear if it correspond to the cases used for training or for testing/validation. This is important, as testing/validation loss curves are the ones that really matter.
2 ) The use of the same symbol "sigma" for the Deblurring and Denoising is confusing. I suggest using something like sigma_b for the deblurring and sigma_d for the denoising or any other way to make them different. For instance in Table 2, 3 and the rest of the paper. (Lines 123, 125..)
3 ) Line 177 - I don´t think that "pruning" is the right word. All the image preparation before the Neural Network is "preprocessing", but pruning has the meaning in the field of making it smaller by removing the unnecessary elements, which is not exactly the case. Same applies for other parts where pruning was used.
4 ) In the discussion, I would talk about the harmonization of images acquired from multiple centers. For instance, here is a reference that could apply to this work:
https://www.sciencedirect.com/science/article/pii/S1053811921009642
5 ) Line 189, 190 - This is a critical point. It is needed to indicate the differences among the images used in the study, to better understand why some specific preprocessing makes it work better. The scanner used is indicated only as "Siemens", but are there different models used in the database? If you combine images from different generations of PET scanners, the differences in resolution and sensitivity of the scanners should be apparent. In that case, the preprocessing used in this work would make them more compatible. If they were all acquired with the same model, maybe the differences between images are caused by the amount of radiotracer used in each specific hospital. But if they all followed a similar procedure, then the motivation or the effect of the preprocessing would be different. So that's the reason why I think it is important to know all that information, as it may have implications for other researchers involved in heterogenous/homogeneous studies.
5 ) Line 266 - "spatial resolution of 91 x 109..." --> a size fo 91 x 109 pixels
6 ) Line 299 - Explain the PSF used. I assume that it is a Gaussian, but this is not the only possible option. Sometimes, a non-uniform PSF is used for deconvolution in PET. So please, indicate it explicitly with the role of the sigma (sigma_b) indicated. Furhtermore, explain why only a 2D PSF was used, instead of a 3D one? At least indicate it in the discussion.
7 ) Line 320 - Use a different sigma here respect to the deblurring one.
8 ) Question for discussion. Is it possible to evaluate both deblurred and denoised images?
9 ) Line 373 - It is not running --> the importance of preprocessing
Author Response
Thank you for review and comment in this manuscript.
We have revised the paper as your suggestion and responded point by point.
Please confirm attached revised manuscript and response files.
Best regards,
Youngjin Lee

Reviewer 2 Report
This is a interesting study evaluating the effects of noise and blurring on 18F-FDG PET images used for predicting Alzjeimer'disease. The authors demonstrated that 18F-FDG PET image preprocessing and cropping improves the explainability and potential clinical applicability of deep learning models.
The methods are accurate and the results are well-presented.
Introduction and discussion are clear.
Minon issues:
- I would suggest to check the manuscript for typo errors (including the title).
- I would suggest to delete the numbers in the abstract because unclear.
Author Response

(The authors gave the same response as above.)

Round 2
Reviewer 1 Report
The authors have successfully addressed all my comments, and suggestions and I do think that now the paper is suitable for publication.